# First Clinical Report of the Intraoperative Macro- and Micro-Photodiagnosis and Photodynamic Therapy Using Talaporfin Sodium for a Patient with Disseminated Lumbar Medulloblastoma

**DOI:** 10.3390/jcm12020432

**Published:** 2023-01-05

**Authors:** Jiro Akimoto, Shinjiro Fukami, Kenta Nagai, Michihiro Kohno

**Affiliations:** 1Department of Neurosurgery, Kohsei Chuo General Hospital, 1-11-7 Mita, Meguro-ku, Tokyo 153-0062, Japan; 2Department of Neurosurgery, Tokyo Medical University, Tokyo 160-0023, Japan

**Keywords:** malignant spinal cord tumor, photodiagnosis, photodynamic therapy, talaporfin sodium

## Abstract

Photodiagnosis (PD) and photodynamic therapy (PDT) using the second-generation photosensitizer talaporfin sodium together with an exciting laser for primary intracranial malignant tumors is well recognized in Japan, and many medical institutions are introducing this new therapeutic option. In particular, intraoperative PDT using talaporfin sodium for infiltrating tumor cells in the cavity walls after the resection of malignant glioma is now covered by health insurance after receiving governmental approvement, and this method has been recommended in therapeutic guidelines for primary malignant brain tumors in Japan. On the other hand, experimental and clinical studies on the development of novel therapeutic strategies for malignant spinal cord tumors have not been reported to date, although their histological features are almost identical to those of intracranial malignant tumors. Therefore, the clinical outcomes of malignant spinal cord tumors have been less favorable than those of malignant brain tumors. In this report, we performed the PD and PDT using talaporfin sodium on a patient with a metastatic lumbar lesion that was detected on magnetic resonance image (MRI) 50 months after the resection of cerebellar medulloblastoma who presented with lumbago and sciatica. We were able to detect the target lesion in the conus medullaris using a surgical microscope, and detected the disseminated medulloblastoma cells floating in the cerebrospinal fluid using a compact fluorescence microscope. Furthermore, we performed PDT to the resected lumbar lesion with the adjuvant platinum-based chemotherapy, and the patient survived a meaningful life for more than 2 years after the lumbar surgery. This report describes the first case of a human patient in whom the efficacy of PD and PDT was demonstrated for a malignant spinal cord tumor.

## 1. Introduction

In Japan, the patients with intracranial malignant brain tumors can receive photodynamic therapy (PDT) using talaporfin sodium with an exciting diode laser under health insurance [1,2,3]. A recent study reported that additional intraoperative PDT for newly diagnosed glioblastoma showed more favorable outcome than conventional treatment only, with a median progression free survival of 19.6 months and median overall survival of 27.4 months, respectively [4,5]. According to these results, medical facilities performing PDT are rapidly increasing in Japan.

Malignant tumors including the glioblastomas arising primarily in the spinal cord are extremely rare and usually the treatments similar to those of malignant brain tumors are performed [6]. Although the treatment results of malignant primary spinal cord tumors are less favorable than those of malignant primary brain tumors because of their location in eloquent areas, research on the development of novel treatment strategies for malignant spinal cord tumors have not been performed to data unlike intracranial tumors [6].

Recently, we encountered a young adult patient with disseminated lumbar medulloblastoma who presented with rapidly progressive cauda equina syndrome, and performed intraoperative photodiagnosis (PD) [7,8,9] and PDT after obtaining informed consent. We hence here report the first clinical case of a patient who demonstrated efficacy of PD and PDT for a malignant spinal cord tumor.

## 2. Clinical Case Presentation

A 24-year-old man without any past medical history presented with a 2-month history of gradually worsening of headache and nausea. A brain CT scan and MRI displayed a heterogeneously enhanced tumor located in the cerebellar vermis extending to the fourth ventricle accompanying the obstructive hydrocephalus. (Figure 1A,B) We performed subtotal tumor resection, but a small amount of tumor tissue invading the floor of the fourth ventricle was left. The histopathological diagnosis was classical type medulloblastoma with apparent Homer-Wright rosettes, and MIB-1 labelling index was 15%. Postoperative MRI confirmed subtotal resection of the tumor without apparent dissemination in the cerebrospinal fluid (CSF). (Figure 1C,D) After surgery, the patient underwent craniospinal radiotherapy of the whole brain of 30 gray (Gy) at 2 Gy/fraction with an additional 20 Gy boost to the posterior fossa and 25.5 Gy irradiation to the whole spine as prophylactic treatment. The patient also concurrently received 4 cycles of platinum-based chemotherapy (cisplatin/etoposide), and the patient was discharged from hospital with independent gait and a Karnofsky performance status score of 80. After his discharge, he got married and began working. However, at a periodic consultation 50 months after surgery, he complained of severe lumbago and sciatica. Brain and spinal MRI displayed nodular enhancement in the conus medullaris and diffuse enhancement in the caudal part of the dura mater without apparent recurrence of the primary tumor, and we suspected CSF dissemination of the medulloblastoma. (Figure 1E,F) While planning the patient’s treatment strategy, his gait disturbance rapidly worsened and urinary incontinence developed owing to cauda equina syndrome. We recommended palliative care, but the patient and his father requested surgery again with the hope that the patient would then be able to go home even for a short period. Therefore, we performed a lumbar surgery with additional PD and PDT after adequate informed consent from the patient and patient’s families. The institutional review board of Tokyo Medical University approved the treatment (study approval no.: No.419).

## 3. Surgery

Twenty-four hours prior to surgery, talaporfin sodium (Laserphyrin^®^, Meiji Seika Pharma Co., Ltd., Tokyo, Japan) was administered as a bolus (40 mg/m^2^) intravenously and the patient was kept in a light-shielded condition under illumination at ≤500 lux. Under general anesthesia, posterior laminoplastic laminotomy of L1 and half laminectomy of Th12 was performed. After midline incision of the dura mater and arachnoid membrane, flowing out of clear CSF was observed. A 5 mL sample of CSF was centrifuged at 3000 rpm for 5 min, and the sediment was collected. A tissue preparation of this sediment was made and the fluorescence from this tissue was assessed using a compact fluorescence microscope in the surgical theater. Under the surgical microscope, a weak fluorescence was detected from the gelatinous tumor tissue with hypertrophic pia mater between several cauda equina nerves, and a small portion of the tumor was resected. (Figure 2A–C) Finally, a single shot of PDT was performed using a 664 nm semiconductor laser (PD laser BT^®^, Meiji Seika Pharma Co., Ltd., Tokyo, Japan) to the conus medullaris through a quartz fiber inside the lens barrel of the surgical microscope with an irradiation power density of 150 mW/m^2^, and an irradiation energy density of 27 J/cm^2^ (irradiation time: 3 min) within a circle (diameter: 1.5 cm) per location. (Figure 2I) Intraoperative assessment using a fluorescence microscope demonstrated that strong fluorescence was detected not only from the cytoplasm of floating tumor cells in the CSF, but also from resected tumor cells with a high nuclear-to-cytoplasmic (N/C) ratio (Figure 2D–H).

## 4. Postoperative Course

The histopathological diagnosis of the lesion in the conus medullaris was malignant small round cell tumor with a high N/C ratio. (Figure 3A,B) After surgery, CSF leakage from the surgical wound continued, and we performed reclosure of the dura mater using perifascial areolar tissue and a subcutaneous fat graft 4 days after the lumbar surgery. The patient’s severe preoperative lumbago gradually improved and he started physical therapy. The patient also started chemotherapy of weekly intrathecal injections using methotrexate (MTX) (6 mg/m^2^) and cytosine arabinoside (Ara C) (20 mg/m^2^), and concurrent oral administration of temozolomide (TMZ) (150 mg/m^2^) per day, 5 days/28 days. Three weeks after the surgery, his sciatica gradually improved and he became able to walk by himself using a T cane. Lumbar MRI displayed a faint enhanced lesion in the conus medullaris, and disappearance of enhancement in the caudal part of the dura mater. Cytology of the CSF also demonstrated improvement from Class III b to Class II. (Figure 3C) As the patient and his family strongly wished for him to be the discharged to go home, intrathecal chemotherapy was terminated after 3 cycles, but temozolomide administration was continued. However, only 2 weeks after discharge, the patient presented with severe neck pain and arm numbness. Brain MRI displayed no apparent recurrence of the primary cerebellar tumor, however, cervical MRI displayed a heterogeneous weakly enhanced tumor with cystic changes in C 3 to C 7 level of the spine. (Figure 3D,E) The patient wished to undergo platinum-based chemotherapy (ifosfamide/cisplatin/etoposide: ICE) in the hospital and then go back to his home. However, respiratory disturbance occurred and as patient and his family did not wish for any life-prolonging treatment, the patient was died 72 months after the first surgery, and 24 months after the lumbar surgery. A post-mortem autopsy was not performed owing to lack of permission from the patient’s family.

## 5. Discussion

In 1980, the first case of PDT performed on a human patient for a malignant brain tumor was reported from Italy [10,11], and thereafter PDT for malignant brain tumors has been performed frequently in many countries for the past 40 years. However, to the best of our knowledge, there has been no case reports to date of PDT performed for a malignant spinal cord tumor. Malignant spinal cord tumors show histological similarities to malignant intracranial tumors [6], and therefore, we expected PDT to show clinically efficacy. However, spinal cord tumors comprise only 2% to 4% of central nervous system tumors, and, in particular, the malignant histology is an extremely rare occurrence [6]. Furthermore, neuro-oncology specialists are not necessarily in charge of the treatment of spinal cord tumors. Owing to these reasons, it is possible that PDT has not been recognized as a treatment option by physicians who usually treat malignant the spinal cord tumors. In fact, according to a recent report from Japan of 1033 cases of intramedullary spinal cord tumors in the previous 12 years [12], there were only 29 cases (2.8%) of tumors demonstrating a malignant histology including WHO grades 3 or 4. Furthermore, in a study of 48 cases during 7 years from France [13], there were only 3 cases (6.3%) of tumors demonstrating malignant histology. Among Japanese cases, after surgery, the adjuvant radiotherapy was performed on 130 patients (12.6%) including those with grade 2 ependymomas, and temozolomide-based chemotherapy was performed on 81 patients (7.8%) [12], however, no other types of treatment were performed in this previous Japanese study [12].

When we search articles in PubMed using the key words “Spinal tumor”, “Spinal cord tumor” and “PDT”, most of the identified articles reported the efficacy and safety of PDT for spinal “vertebral” metastatic tumors as preclinical and clinical research [13,14,15,16,17]. However, there were no articles analyzing the efficacy of PDT for spinal “cord” tumors. Therefore, the case presented in this article is the first case of a human patient who received additional intraoperative PDT for a malignant spinal cord tumor. Several reports have documented the utility of the fluorescence-guided resection (FGR) using a photosensitizer for the treatment of spinal cord tumors. The main photosensitizer used for spinal cord tumor is 5-aminolevulinic acid (5-ALA), and FGR is used like as for brain tumors, and fluorescence sodium was used in an article [18,19,20,21,22,23,24,25,26,27]. Interestingly, many reports have described the utility of FGR using 5-ALA for the resection of intramedullary ependymomas WHO grade 2 [18,19], which do not usually demonstrate fluorescence of the 5-ALA metabolite protoporphyrin IX (PPIX) if the tumor occurred in the brain. Wainwright et al. [18] evaluated the intraoperative fluorescence emitted from PPIX using 5-ALA to detect spinal dissemination from malignant brain tumors, and they reported that disseminated glioblastomas usually demonstrated PPIX fluorescence, however, disseminated medulloblastomas and choroid plexus papilloma did not demonstrated fluorescence of PPIX [18]. In the present study, not only the disseminated medulloblastoma tissue in the conus medullaris but also the floating tumor cells in the CSF demonstrated positive fluorescence using talaporfin sodium, using our previously reported method [7,9]. Although, the mechanism of the diffusion of intravenously administered talaporfin sodium to the CSF and its uptake into the disseminated tumor cells via the enhanced permeability and retention effect is straightforward, this is the first report to date of the intraoperative detection of fluorescence from floating tumor cells in the CSF in clinical practice [9]. Our previously reported method regarding intraoperative fluorescence cytology in the CSF has many clinical implications for the management of malignant central nervous system tumors [9].

For the management of disseminated adult medulloblastoma in the CSF, the NCCN guideline 2022 recommend craniospinal irradiation and high-dose chemotherapy using cyclophosphamide and etoposide or temozolomide [28,29,30,31,32]. In addition, although the efficacy has not been confirmed, the administration of vismodegib for patients with genetic mutations in the Sonic hedgehog pathway has been recommended [28]. In the present case, whole spinal irradiation had already been performed as a preventative treatment during the initial tumor management. Therefore, after lumbar surgery with PD and PDT, we performed the intrathecal injection of MTX with Ara C, and the oral administration of temozolomide as an adjuvant chemotherapy [29,30,31,32]. However, owing to early cervical intramedullary recurrence, we performed intense chemotherapy using ICE regimen, which has been demonstrated to be effective for childhood medulloblastoma patients in Japan [33,34]. As a result, the patient was in a stable condition for 2 years after the lumbar surgery and survived for 6 years after the initial treatment, and he was also able to have a son by artificial insemination using the sperm that he cryopreserved before the chemotherapy.

## 6. Conclusions

We reported a case of an adult patient with classical type of medulloblastoma who demonstrated the CSF dissemination and distant recurrence in the conus medullaris 50 months after the initial treatment. In particular, we reported the utility of intraoperative macro- and micro-PD using the photosensitizer talaporfin sodium for this complicated condition, and performed PDT to the metastasized lumbar nodule. There were no adverse events directly associated this PDT, and optimal maintenance systemic chemotherapy after surgery resulted in a satisfactory clinical course of 2 years of survival after the PDT. This case was the first human patient with a disseminated malignant spinal cord tumor in whom intraoperative PD and PDT was performed using a photosensitizer with an exciting laser. We hope that spread of the recognition of the clinical utility of this method for malignant spinal cord tumors, will contribute towards improvement of therapies for this severe condition.

## Figures and Tables

**Figure 1 jcm-12-00432-f001:**
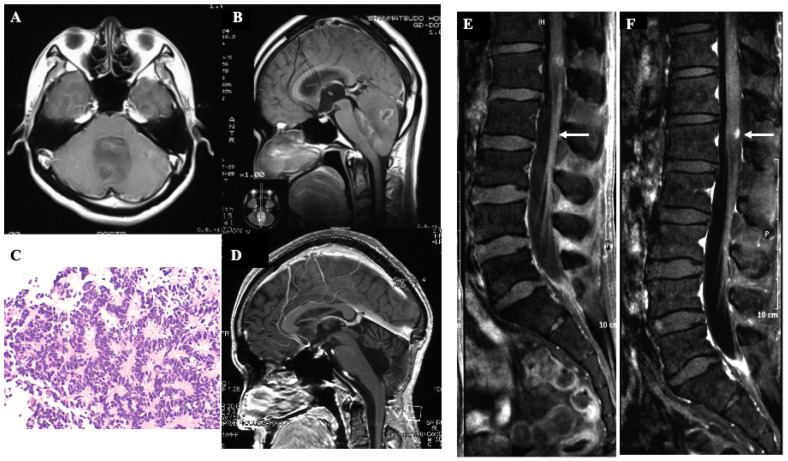
Radio-pathological findings of the patient before the lumbar surgery. (**A**,**B**): Preoperative gadolinium-enhanced T1-weighted MRI images of the brain displayed a heterogeneously enhanced tumor in the cerebellar vermis accompanied with obstructive hydrocephalus. (**C**): The histopathological diagnosis of the tumor was a classic type medulloblastoma with abundant typical Homer-wright rosette in the hematoxylin and eosin stain (×100). (**D**): Postoperative gadolinium-enhance T1-wegihted MRI image of brain displayed subtotal resection of the cerebellar tumor. (**E**,**F**): Gadolinium-enhanced T1-weighted image of lumbar spine displayed a nodular enhanced mass ((**F**): white arrow) with enhancement of meninges of cauda equina ((**E**): white arrow).

**Figure 2 jcm-12-00432-f002:**
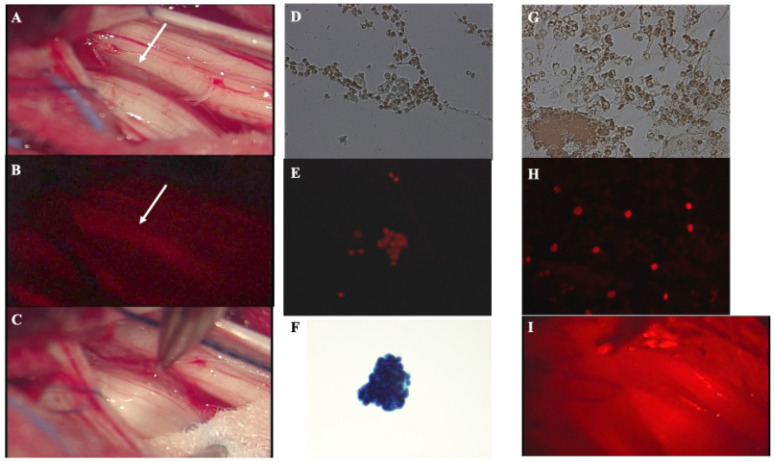
Intraoperative findings of the lumbar surgery and practice of PD and PDT for the tumor located in the conus medullaris. (**A**): Intraoperative finding of an intradural lumbar lesion demonstrating a gelatinous pink tumor located in the conus medullaris between the cauda equina (white arrow). (**B**): Intraoperative PD using a surgical microscope demonstrated weak fluorescencefrom the tumor (white arrow). (**C**): Piece by piece resection of the tumor. (**D**,**G**): Intraoperative fluorescence cytology of sediment from the CSF (**D**) and resected tumor tissue (**G**) under white light demonstrated a cluster of tumor cells. (**E**,**H**): Intraoperative fluorescence cytology of the sediment from the CSF (**E**) and resected tumor tissue (**H**) demonstrated strong fluorescence of talaporfin sodium in the cytoplasm of the cluster of tumor cells with a high N/C ratio. (**F**): Giemsa stain (×400) of the sediment from the CSF demonstrated a cluster of tumor cells. (**I**): We performed a single shot of irradiation of PDT to the conus medullaris. The magnification of histological images: (**D**,**E**,**G**): ×200, (**F**,**H**): ×400.

**Figure 3 jcm-12-00432-f003:**
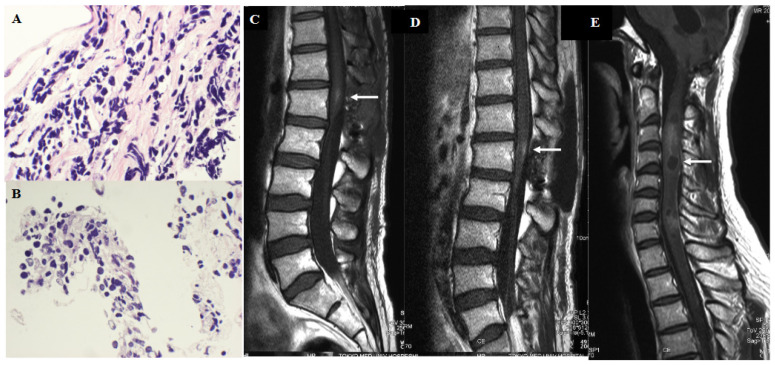
Radio-pathological course of the patient after the lumbar surgery. (**A**,**B**) Histopathological findings of the resected tumor in the conus medullaris demonstrated a cluster of small round cell tumor cells with a high N/C ratio (×400). (**C**): Postoperative gadolinium-enhanced T1-weighted image of the lumbar spine displayed a faint tumor nodule (white arrow) in the conus medullaris without enhancement of the meninges of the cauda equina. (**D**,**E**) Gadolinium-enhanced T1-weighted MRI of the lumbar spine (**D**) and cervical spine (**E**) taken 2 months after the lumbar surgery. The lumbar lesion displayed faint tumor enhancement ((**D**): white arrow), and the heterogeneous enhanced intramedullary mass was located between the C3 and C7 levels ((**E**): white arrow).

## Data Availability

The datasets generated and/or analyzed during the current study are available from the corresponding author on reasonable request.

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
