# Peer review of "First Clinical Report of the Intraoperative Macro- and Micro-Photodiagnosis and Photodynamic Therapy Using Talaporfin Sodium for a Patient with Disseminated Lumbar Medulloblastoma"

_jcm, 2023, doi:10.3390/jcm12020432_

Round 1
Reviewer 1 Report
The manuscript "First clinical report of the intraoperative macro and micro photodiagnosis and photodynamic therapy using talaporfin sodium for a patient with disseminated lumbar medulloblastoma" by Akimoto et al. presents the first clinical case of PD and PDT on the patient with disseminated lumbar dedulloblastoma.
The paper will be interesting for medical community after some amendments.
Comments.
Can please the authors provide more information about talaporfin sodium-based PDT clinical protocol. Was the concentration of talaporfin sodium bolus injection 2mg/kg (as for esophageal cancer). The authors indicate the light dose and light flurence rate. What was the irradiation time? According to calculation it was too long. Please provide the illumination time and in addition, how irradiation was done, with fibers? What kind of them?
The authors note in the Surgery section that a tissue preparation was made from CSF and fluorescence from this tissue was further assessed. Please precise how the tissue preparation was made.
What was the reason to leave a small amount of tissue invading the floor of the 4th ventricle?
Please comment on that.
Author Response
# The paper will be interesting for medical community after some amendments.
We appreciate your very positive comments. We will examine your comments, revise this paper, and resubmit it as appropriate for this journal.
# Can please the author provide more information about talaporfin sodium-based PDT clinical protocol. Was the concentration of talaporfin sodium bolus injection 2mg/kg(as for esophageal cancer). The authors indicate the light dose and light fluence rate. What was the irradiation time? According to calculation it was too long. Please provide the illumination time and in addition, how irradiation was done, with fibers? What kind of them?
Thank you for your precise comments. The protocol described in this paper, talaporfin-based PDT for malignant central nervous system tumors, has already been approved by the Japanese government and is covered by insurance. In the surgery section of this paper, we described the specific dosage of bolus injection of talaporfin sodium is 40 mg/m2. And, we described the laser irradiation conditions as power density of 150 mW/cm2 and energy density of 27 J/cm2. However, I think the reviewer‘s comment is correct that there was insufficient description of what methodology was used to laser irradiation to the target. The methodology of this treatment was developed by the first author and has already been described in detail in reference paper 1, but we have decided to describe the methodology in more detail in revised paper.
# The authors note in the surgery section that a tissue preparation was made from CSF and fluorescence from this tissue was further assessed. Please precise how the tissue preparation was made.
Thank you for your interest in our intraoperative fluorescent cytology system.  This methodology was developed by the first author and has already been described in detail in reference paper 9. As described in the surgery section, aspirated CSF is centrifuged at 3000 rpm for 5 minutes, the sediment is smeared on a nonfluorescent glass slide, fixed with alcohol, and inserted into a compact fluorescence microscope in the operating room to catch up the fluorescence of talaporfin sodium using a special filter. The first author has called photo-diagnosis observed under an operating microscope "macro photo-diagnosis" and has applied it in many brain tumor surgeries. This has already been reported in reference paper 7. On the other hand, the method of capturing fluorescence of talaporfin sodium emitted from tumor cells using a compact fluorescence microscope during the surgery is called micro photo-diagnosis in reference paper 9. We hope you will understand this point.
# What was the reason to leave a small amount of tissue invading the floor of the 4th ventricle?
Thank you for your valuable comments. As an intraoperative finding at the time of the initial surgery in this case, the tumor tissue was firmly attached to the floor of the fourth ventricle, and we surmised that there was a high risk of damaging the floor of the fourth ventricle by removing this area. Injury to this area would have a risk of affecting swallowing and speech functions, and would significantly worsen the patient's postoperative performance status. Therefore, we decided that we had no choice but to leave the tumor in this region.

Reviewer 2 Report
This manuscript produced by Akimoto et. al., reports a surgical trial performed on a human patient, of which involved incision of a human spine, administering a second-generation photosenitizer, and deactivating cancer cells using Photodynamic therapy (PDT). Although the topic presented is attractive and of interested a wide community, this paper reports only the single case of medicine practice and general observations of the patient during the pre- and post- operations. Hence, no quantitative or qualitative research/analysis was performed in this study.
Please find a number of major concerns also presented in this manuscript are summarised as follow:
1. It is unclear if there is any ethical approval and patient's consent was considered for this trial and for the intension of publication. This paper has unprofessionally reported patient's name (page 2), which must be omitted in any scientific publication. This paper also provided excessive information about the patient's personal life, which appeared to be unnecessary for the scientific study.
2. It is unclear if any of the listed authors is/are the lead surgeon who performed the surgery, as this is not mentioned in the Author contribution section. Hence there is an authorship conflict and again ethical concern here.
3. It is important to check if the authors have correctly claimed to use Talaporfin Sodium (TS) for photodiagnosis purpose. To the best of my knowledge, without using additional fluorescent reagent, the cells cannot be visible using TS alone. The 664nm wavelength is only for activating the ROS process via the PDT technology.
4. It is also questionable that the authors claim to have use the PD brand semiconductor laser for the irradiation step and not the optical fibre during the surgical procedure. It is known that such procedure requires the emitted wavelength to penetrate cells for effective results. If wrong equipment was used during the clinical trial, here is another professional and ethical issues raised.
Author Response
#Although the topic presented is attractive and of interested a wide community, this paper reports only the single case of medical practice and general observations of the patient during the pre- and post-operations. Hence, no quantitative or qualitative research/analysis was performed in this study.
We sincerely appreciate the high evaluation of this paper. At the same time, it was also pointed out to us that this paper is merely a case report and does not provide sufficient qualitative and quantitative scientific analysis. I understand that the reviewer is exactly right. However, we believe that this is the first paper in the world to demonstrate the usefulness of intraoperative photo-diagnosis and photodynamic therapy using talaporfin sodium for malignant tumors of the spinal cord. We wanted to publish this world-first finding in this prestigious journal. We hope you will understand this point.
#1 It is unclear if there is any ethical approval and patient’s consent was considered for this trial and for the intension of publication. This paper has unprofessionally reported patient’s name (page2), which must be omitted in any scientific publication. This paper also provided excessive information about the patient’s personal life, which appeared to be unnecessary for the scientific study.
Thank you for pointing out a very important ethical point. Regarding the patient's consent, we have obtained consent from the patient at the time of selecting the treatment for the tumor disseminated in the lumbar spinal cord. Most of the certificates are written in Japanese, but there are some that are handwritten by the patient. In addition, since the patient has already passed away, we contacted his wife again to obtain her written consent for the submission of this paper. This document is attached in PDF format. We apologize for the description that went into the patient's personal information. We will erase the fact that he is a photographer in the revised version. However, we would like to ask your permission to mention in the last sentence of the discussion part that the patient received his first son through artificial insemination, which we believe indicates the success of our treatment.  The reviewer stated that the patient's name is listed on page 2, but perhaps he misunderstood Chang's stage, which is used to evaluate the clinical stage of medulloblastoma. Chang staging system is widely accepted in brain oncology around the world as an evaluation of the clinical stage of medulloblastoma. However, recent advances in molecular biological analysis have led to advances in genetic classification in medulloblastoma, and staging is rarely done on the clinical feature. Therefore, in light of the reviewer's misunderstanding, we have decided to remove Chang's staging from this paper. Please check it out at revised version.
#2 It is unclear if any of the listed authors is/are the lead surgeon who performed the surgery, as this is not mentioned in the author contribution section. Hence there is an authorship conflict and again ethical here.
Thank you for pointing out this important point. The first author was in charge of all surgeries and treatments for this patient. Then the 2nd and 3rd authors, together with the first author, managed the entire clinical course of the patient. As the authorship, the last author is the author's supervisor and is responsible for the final review of this paper. This point is added in the revised version.
#3 It is important to check if authors have correctly claimed to use talaporfin sodium (TS) for photodiagnosis purpose. To the best of my knowledge, without using additional fluorescent reagent, the cells cannot visible using TS alone. The 664nm wavelength is only for activating the ROS process via the PDT technology.
Thank you for your interest in the intraoperative real-time visualization of fluorescence from tumor cells developed by the first author. We describe this methodology in detail in reference paper 9 and show our experience with 25 brain tumor cases. We believe that the fluorescence of talaporfin sodium can be reliably captured by using a special filter that catches up the 672 nm fluorescence emitted by irradiating excitation light to talaporfin sodium incorporated into the lysosomes of tumor cells. If you wish, we can provide a PDF file of reference paper 9.
#4 It is also questionable that the authors claim to have use the PD brand semiconductor laser for the irradiation step and not the optical fiber during the surgical procedure. It is known that such procedure requires the emitted wavelength to penetrate cells for effective result. If wrong equipment was used during the clinical trial, here is another professional and ethical issues raised.
We appreciate your questions regarding the details of the PDT methodology. We acknowledge the lack of methodological details as you mentioned, and have added a bit more detail on the laser irradiation method in the revised version. Thank you very much for your consideration. Also, as mentioned in the introduction part, this methodology was developed by the first author and has already been approved by the Japanese government through a physician-led clinical trial. The laser equipment we use is also approved by international electrical standards, and we believe that there are no credibility or ethical issues in our PDT thinking. Thank you in advance for your consideration.
